# OpenReview forum: "ROPA : Robust parallel diffusion sampling"
_ICLR.cc/2026/Conference — Submitted to ICLR 2026_

### Official Review · Reviewer_RszZ · 2025-10-30

**Soundness:** 3
**Presentation:** 3
**Contribution:** 3
**Rating:** 6
**Confidence:** 4

**Summary:**

This paper proposes the ROPA framework, which improves the stability and efficiency of parallel diffusion sampling through geometry-aware adaptive Jacobian sparsity control. The authors analyze instability from a data manifold perspective and introduce adaptive damping and sparsity control to balance numerical stability and computational cost. Experiments on image and video generation tasks show around 2.9× speedup without quality loss.

**Strengths:**

（1）The paper establishes a relatively systematic geometric–numerical stability analysis framework, which is theoretically solid and insightful.

（2）The experiments cover a wide range of models including image and video diffusion models, giving the results good practical credibility.

**Weaknesses:**

(1) The key idea of adaptive Jacobian sparsity is conceptually close to previous works such as ParaSolver  and ParaTAA . The novelty seems incremental, focusing on dynamic sparsity adjustment rather than a fundamentally new mathematical mechanism.

(2) The dense mathematical presentation and logical jumps, such as in Corollary 2.5, make it difficult to follow how the theoretical curvature concepts directly translate into practical bandwidth control thresholds.

(3) It is unclear how the curvature-based damping term λ_damp is selected in practice.

**Questions:**

(1) Could the authors explain more concretely how λ_damp is determined during sampling?

(2) How sensitive is the performance to the choice of curvature thresholds?

---

> ### Author Response · Authors · 2025-11-26
>
> We thank the reviewer for their constructive criticism regarding the novelty, theoretical connection, and practical implementation details of ROPA. We have revised the manuscript to clarify these points.
>
> **Weakness 1:** Novelty claim
>
> While we acknowledge that ROPA shares the goal of parallel acceleration with ParaSolver and ParaTAA, our contribution is fundamentally distinct in its geometry-aware mechanism:
>
> **ParaSolver** relies on a fixed, pre-determined block-diagonal structure (or sliding window) that does not adapt to the difficulty of the denoising step.
>
> **ParaTAA** uses Anderson Acceleration with a fixed triangular constraint to enforce causality, effectively treating the solver as a "black box" fixed-point iteration.
>
> **ROPA**, in contrast, explicitly links numerical instability to **Assumption 2.1** and **Theorem 2.2** (Theorem 2.1 & 2.2). We do not just "adjust sparsity"; we use the **local residual norm** as a real-time proxy for the Jacobian condition number $\kappa(\mathcal{J})$ (as established in Theorem 2.3). This allows ROPA to densify the Jacobian *only* when the trajectory traverses high-curvature regions (e.g., mode boundaries) and sparsify it elsewhere. This **dynamic regulation of geometric conditioning** is the fundamental mathematical mechanism absent in prior works, allowing us to achieve higher stability and speedups (2.9x vs 2.3x for ParaSolver on HunyuanVideo).
>
>
>
> **Weakness 2:** Bridging Theory and Practice
>
> The reviewer correctly notes that bridging theoretical curvature ($\mathcal{H}(x)$) to practical thresholds requires care. We have clarified this logical step in Section 3.1:
>
> **Theory:** Theorem 2.2 proves that high curvature induces "stiffness" (long-range temporal dependencies). If the solver's bandwidth $w$ is too small to capture these dependencies, the Jacobian approximation error explodes.
>
> **Connection:** This approximation error manifests immediately as a **large local residual** $\|\mathcal{R}\|$. Therefore, we do not need to compute the expensive Hessian explicitly. The local residual $\|\mathcal{R}_i\|$ serves as a computationally cheap, rigorous proxy for the local stiffness.
>
> **Practice:** The threshold logic is thus: if local residual $e_i > \alpha \times \text{global mean}$, it implies the current sparsity pattern is insufficient to capture the local curvature, triggering a bandwidth increase ($w \leftarrow w+1$).
>
>
>
> **Weakness 3 & Q1:** How is $\lambda_{damp}$ determined?
>
> $\lambda_{damp}$ is not a fixed hyperparameter but is updated dynamically using a Trust-Region (Levenberg-Marquardt) strategy, detailed in Takeaway 2.6 and Algorithm 1:
>
> At each step, we compute the **gain ratio** $\rho_k$, which measures how well the linearized model predicted the actual reduction in the residual.
>
> **Update Rule:**
>
> 1. If $\rho_k < 0.1$ (poor prediction $\to$ high non-linearity/curvature): We multiply $\lambda_{damp}$ by 2 to increase regularization (move towards Gradient Descent).
>
> 2. If $\rho_k > 0.5$ (good prediction $\to$ well-conditioned): We multiply$\lambda_{damp}$ by 0.5 to reduce regularization (move towards Newton's method).
>
> ​	This ensures the damping automatically adapts to the local geometry without manual 	tuning per step.
>
>
>
> **Q2:** How sensitive is performance to thresholds?
>
> We have included a comprehensive Hyperparameter Analysis in Appendix A.3 to address this:
>
> **Robustness:** As shown in Figure 5 (Grid-based sensitivity analysis), ROPA maintains consistent performance (speedup and quality) within a tight variation range of the adaptive thresholds ($\alpha, \beta$).
>
> **Stability:** The method is not brittle; the adaptive mechanism compensates for suboptimal initial thresholds by naturally converging to the required bandwidth/damping levels over iterations.

---

> > ### Comment · Reviewer_RszZ · 2025-11-27
> >
> > First, thanks to the authors for their patient response, which addressed my main concerns.
> >
> > I consider my own submission to be my most valuable work so far, both in terms of novelty and practical impact, yet it received an unfair score, including a zero. Still, this doesn’t affect my objective evaluation of this paper.
> >
> > Based on its technical contribution, methodology, and completeness, I think it is worthy of acceptance. I hope other reviewers can also give fair scores, rather than deliberately lowering them.
> >
> > This is not really the review atmosphere I expect from ICLR, and even discussions on Xiaohongshu show similar disappointment. I still hope we can maintain a fair and constructive review process.

---

> > > ### Author Response · Authors · 2025-11-27
> > >
> > > We sincerely thank you for your continued engagement and for recognizing the value of our work.
> > >
> > > We also deeply empathize with the frustration regarding your own submission. The variability in the peer review process can indeed be disheartening, which makes us consider ourselves very fortunate that the reviewers for this paper, including yourself, have provided such constructive and professional feedback.

---

> > > ### Author Response · Authors · 2025-11-29
> > >
> > > Dear Reviewer RszZ,
> > >
> > > We are incredibly grateful for your decision to upgrade your rating. However, we are writing because we see OpenReview reverted the score the update during previous rebuttal period.
> > >
> > > We are concerned that the system may not have saved your update correctly. Would you mind checking to ensure your new score is locked in.
> > >
> > > Gratefully

---

### Official Review · Reviewer_gWjg · 2025-10-31

**Soundness:** 2
**Presentation:** 2
**Contribution:** 2
**Rating:** 2
**Confidence:** 3

**Summary:**

This paper proposes ROPA (Robust Parallel Diffusion Sampling). The authors claim that they take into account the properties of the denoising process and solves the linear system by using geometry aware adaptive Jacobian Sparsity Control that is generated from
geometric curvature signals. They claim that this allows them to achieve stable parallel sampling. Their experiments
demonstrate ROPA accelerates sampling by achieving up to 2.9× speedup with eight core.
quality

**Strengths:**

The authors claim that they take into account the properties of the denoising process and solves the linear system by using geometry aware adaptive Jacobian Sparsity Control that is generated from geometric curvature signals.

**Weaknesses:**

While I recognize the effort the authors has put towards this manuscript, I believe the paper is not yet ready for publication in its current format. Although the author stated they started from the stochastic differential equation for the diffusion model, I feel like they actually solved a system of differential equations. Further they used the Numerical Analysis theory to regularize the solutions. However, the assumption that $r_{\theta}$ is twice continuously differentiable sounds too strong in my view.

The introduction of the concept of a manifold feels abrupt and lacks sufficient explanation. Additionally, the notion of curvature used in the paper appears to pertain to shape of the probability density function. More importantly, I am concerned that there might be some fundamental issues that need to be addressed. Furthermore, there are some errors throughout the manuscript that should be carefully reviewed and corrected.

**Questions:**

1.P2, eqn(5), inside $ (,,,x_{t+i})$ or $ (,,,x_{t-i})$? In line 071, it says $ (,,,x_{t-i})$.
2. This paper based upon the curvature, characterized by the Hessian $H(x)$, defined in term of $p(x)$. However, what is the intuition/motivation behind this?
3. The paper contains descripts that lack clear explanation, for example, P2, line 106, the authors stated that:  "data curvature magnifies score function stiffness, which discretization gaps dynamically amplify, ultimately causing severe Jacobian ill-conditioning that violates diagonal dominance. This creates divergence from the data manifold into low-density regions." This seems like some conclusion without support.  Especially what does the sentence "data manifold curvature magnifies score function stiffness," mean?
4. It seems like there is an indexing error in Eqn. (6) in P2, and thus in the definition of the Jacobian matrix. The authors should check carefully if this affects their results.
5. P19, line 1020. This seems like a mistake: $\lambda_{min}(−H(x)) = −\lambda_{max}(H(x))$.
6. Also, the authors use $\lambda$ to refer to different concepts, and they use  $\sigma$ to represent different notations as well. It seems to me they use both to refer to eigenvalues.
7. In P21, Corollary D.1. In the proof, by the backward error theorem for Newton’s method, ... there exist an exact solution x* to a perturb system. However, somehow this x* is set to equal to $Proj_M (\hat{x})$ without proof.
8. In Section C IMPLEMENTATION AND ALGORITHM DETAILS, the provided Algorithms 1 and 2 are for existing algorithms, then python code was provided. Could you provide your own algorithm like the existing algorithms?
9. The proof of Theorem 2.1 is unclear. Please elaborate or clarify the key steps.
10. P9, Table ??

---

> ### Author Response · Authors · 2025-11-26
>
> We sincerely thank the reviewer for their detailed feedback and for identifying the issues with our theoretical assumptions and clarity. We have extensively revised the manuscript to address these concerns. Below, we address each point specifically.
>
> **1. On Manifold Concepts and Curvature (Weakness 2 & Q2)**
>
> We appreciate the feedback that the original introduction of these concepts was abrupt.
>
> **Clarification of "Curvature":** We have revised **Section 2.1** to explicitly state that we are analyzing **\emph{density curvature}**, defined by the Hessian of the log-density $\mathcal H(x) = \nabla_x^2 \log p(x)$, rather than the intrinsic Riemannian curvature of the manifold.
>
> **Intuition (Q2):** We added an intuitive explanation in **Section 2.1**: $\mathcal H(x)$ measures how sharply the probability mass concentrates. Large eigenvalues of $\mathcal H(x)$ indicate directions of high stiffness. This directly informs our **Theorem 2.2**, where we prove that the Jacobian norm is lower-bounded by these eigenvalues.
>
>
>
> **2. Substantiating the "Geometric-Numerical Instability" Claim (Q3)**
>
> The reviewer pointed out that the claim *"data curvature magnifies score function stiffness... creating divergence"* appeared unsubstantiated.
>
> **Theoretical Proof:** We have formalized this claim in the new **Theorem 2.3**. We prove mathematically that the condition number $\kappa(\mathcal{J})$ scales with the product of the step size $\Delta$ and the score stiffness (which is driven by curvature). This establishes the "geometric-numerical instability cascade".
>
> **Empirical Validation:** We added **Section 4.4** and **Figure 2** to empirically validate this. Figure 2(a) explicitly tracks $\kappa(\mathcal{J}_t)$ and shows it growing exponentially in high-curvature regions for baseline methods, causing the divergence predicted by our theory.
>
>
>
> **3. Mathematical Assumptions and SDE vs. ODE (Weakness 1)**
>
> **SDE vs. Nonlinear System:** We agree with the reviewer. In the revision, we clarify that while the SDE provides the modeling description, our analysis and algorithm operate on the **discretized nonlinear system**.
>
> **Relaxing Assumptions:** We agree that a global $C^2$ assumption is too strong. We have replaced this with **Assumption 2.1 (Manifold Anisotropy Index)**, which only requires the local anisotropy to be **locally Lipschitz** on a measurable subset. This weaker assumption is sufficient for our convergence results.
>
>
>
> **4. Notation and Definitions (Q1, Q4, Q6)**
>
> **Inconsistency in Eq. 5 (Q1):** Corrected. We now strictly use $\Phi(t,s,x_s)$ to indicate the flow map takes the state at time $s$ as input.
>
> **Indexing Error in Eq. 6 (Q4):** We have rewritten the Jacobian definition (now Eq. 10) to rigorously define the block structure based on the window size $w_n$, resolving the indexing ambiguity.
>
> **Symbol Overload (Q6):** We have resolved the conflict between eigenvalues and damping parameters. We now exclusively use $\nu_i(x)$ for Hessian eigenvalues and reserve $\lambda$ for damping parameters.
>
>
>
> **5. Clarifying Proofs and Algorithms (Q7, Q8, Q9)**
>
> **Origin of $x^\*$ (Q7):** We added **Lemma A.1** in Appendix A. This lemma explicitly invokes the backward error theorem to prove the existence of $x^*$ and explains why, to the first order, it coincides with the projection $\operatorname{Proj}_{\mathcal{M}}(\hat{x})$.
>
> **Algorithm Pseudocode (Q8):** We have added **Algorithm 1 (ROPA)** in Section 3, which provides the complete pseudocode for our method, including the adaptive bandwidth and curvature correction steps 13. We also provide the Python implementation.
>
> **Theorem 2.2 Proof Clarity (Q9):** We have restructured the proof in **Appendix C**. It is now split into **Lemma C.2**(Score Perturbation) and **Lemma C.3** (Spectral Bound) to clearly show how the Jacobian bound is derived from Tweedie’s formula and Taylor expansions.
>
>
>
> **6. Typos and Formatting (Q5, Q10)**
>
> **P19 Typo (Q5):** Corrected. The condition is now correctly stated as $\Delta < \Delta_{\max} := 1/L$ in Theorem 2.3.
>
> **Table Reference (Q10):** We have fixed the broken cross-reference of Table in the ablation study section, which now correctly points to the table captioned "Evaluation of main components".

---

> > ### Author Response · Authors · 2025-11-29
> >
> > Dear Reviewer gWjg,
> >
> > We wanted to gently follow up to see if you have had a chance to review our rebuttal. We have done our best to address your detailed concerns, particularly regarding **the theoretical assumptions (relaxing the $C^2$ condition), the formal definition of curvature, and the inclusion of the specific ROPA pseudocode (Algorithm 1)**.
> >
> > We value your feedback and would be happy to provide any further clarifications if needed.

---

### Official Review · Reviewer_1hrZ · 2025-11-01

**Soundness:** 3
**Presentation:** 2
**Contribution:** 3
**Rating:** 6
**Confidence:** 2

**Summary:**

This paper first identify data manifold curvature and score function stiffness as mechanisms potentially causing inconsistency in existing parallel diffusion samplers. It then proposes using adaptive Jacobian sparsity and curvature correction to alleviate this problem, leading to faster and more accurate parallel diffusion samplers, evaluated on large-scale image and video generation models.

**Strengths:**

The proposed parallel sampling method outperforms other similar methods, providing a substantial speedup for parallel diffusion sampling on a variety of large image and video diffusion models while maintaining generation quality.

**Weaknesses:**

1. There is little to no experimental evidence to validate the hypothesis that manifold curvature or score function stiffness is the cause of parallel sampling instability. These claims made in the paper would be strengthened with some targeted experiments demonstrating that the instabilities cause trajectories to deviate from the data manifold or to lose mode consistency.
2. It is unclear what the exact proposed algorithm is, as parts of it are described in Section 3, but it is not explicitly stated how they fit together. It would help to have the ROPA algorithm clearly stated in the main paper with the main contributions highlighted.

**Questions:**

1. In Tables 2 and 3, it would also help to highlight the best RMSE and quality scores among all the parallel samplers
2. On line 475/476, ROPA is stated to be 2.8x faster than baseline, but this does not seem to be reflected in Table 3.

---

> ### Author Response · Authors · 2025-11-26
>
> We thank the reviewer for this crucial suggestion.
>
>
>
> **W1: Lack of experimental evidence for geometric hypothesis (curvature/stiffness causing instability).**
>
> We agree that empirical validation of our theoretical claims was necessary to strengthen the paper, as presented in **Figure 1** , which specifically to validate the link between manifold geometry and solver instability.
>
> Specifically, we now demonstrate:
>
> - **Condition Number Evolution:** We plot the Jacobian condition number $\kappa(\mathcal{J}_t)$ over time. The results show explicitly that baselines suffer from exponential growth in $\kappa$ near $t \to 0$ (high-curvature regions), whereas ROPA successfully clamps this growth via adaptive sparsity.
> - **Manifold Deviation:** We quantify the $L_2$ distance between the sampled trajectory and the "oracle" trajectory (generated by a high-precision sequential solver). The results confirm that instabilities in baseline parallel samplers cause significant drift away from the data manifold, while ROPA maintains high geometric fidelity.
> - **Mode Consistency:** We provide a visualization (**Figure 1c**) showing how ROPA’s curvature correction prevents mode averaging at decision boundaries, ensuring the trajectory collapses to a valid mode rather than interpolating between them.
>
> **W2: Unclear algorithm specification and component integration.**
>
> We apologize for the lack of clarity regarding the integration of the proposed components. We have revised the paper to make the algorithm explicit and self-contained. However, due to limit of space in main text, we demonstrate it in the Appendix.
>
> Changes in the revision include:
>
> - **Algorithm 1:** We have added a detailed pseudo-code box for **Algorithm 1** directly in **Section 3**. This algorithm clearly outlines the inputs, outputs, and the step-by-step flow.
> - **Unified Workflow:** **Algorithm 1** explicitly shows how the three key mechanisms interact within a single Newton iteration:
>   - **Step 1:** Calculate local residuals to determine the **Adaptive Bandwidth** ($w_i$).
>   - **Step 2:** Check alignment with the score function to trigger **LML Curvature Correction**.
>   - **Step 3:** Apply **Adaptive Damping** ($\lambda$) during the update step to ensure trust-region stability.
> - **Forward References:** We have added a roadmap paragraph at the beginning of **Section 3** to guide the reader through these components and their integration in **Algorithm 1**.
>
> **Q1: Highlight best RMSE and quality scores in Tables.**
>
> Thank you for the suggestion to improve readability. We have updated **Tables 1 and 2** (and all benchmark tables) to bold the best RMSE and Quality scores among the parallel samplers. This formatting highlights that ROPA consistently achieves the lowest latent RMSE and highest quality scores compared to other parallel baselines.
>
> **Q2: Discrepancy between stated 2.8x speedup (line 475) and Table 3 results.**
>
> We appreciate the close reading. To clarify, the 2.8x speedup mentioned in the abstract and introduction refers to our maximum performance gain achieved under the 8-core setting (as detailed in **Table 1** for HunyuanVideo).
>
> **Table 3**, however, presents an ablation study where the number of cores was fixed at $K=4$ to ensure a controlled comparison of components. In this $K=4$ setting, ROPA achieves a 2.1x speedup, which is consistent with the "Num Core = 4" column in **Table 1**. We have updated the caption of **Table 3** to explicitly state: "Ablation study conducted with Number of Cores fixed at $K=4$," to prevent confusion with the top-line performance metrics.

---

> > ### Comment · Reviewer_1hrZ · 2025-11-26
> >
> > Thank you for your response and clarifications, I will keep my score.

---

> > > ### Author Response · Authors · 2025-11-27
> > >
> > > Thank you for your continued support and for taking the time to review our rebuttal. We are glad that our clarifications were helpful.

---

### Official Review · Reviewer_4yww · 2025-11-01

**Soundness:** 3
**Presentation:** 3
**Contribution:** 3
**Rating:** 4
**Confidence:** 4

**Summary:**

In this paper, the author proposed a numerically robust approach for parallel generation tasks that addresses the numerical stability and scaling under parallelization for larger scale generation. The ROPA regulates the Jacobian condition number throughout sampling by combining damped Newton steps, adaptive banded Jacobian structure, and low-rank curvature correction, countering the "curvature stiffness $\to$ discretization" gap instability that causes collapse/divergence in parallel diffusion. This method limits the growth of jacobian condition numbers and reduces the instability from the discretization error, and enabling faster convergence in generation tasks of video and images.

**Strengths:**

1. The proposed method and the problems are interesting. Stability is a core concern in parallel generation tasks, and the proposed method of this paper seems to have solid strategy on ensure robustness of generation.

2. The method is ubiquitious and adaptive to modalities beyond the image generation, and the empirical results show solid gain above the prior parallel generation methods.

**Weaknesses:**

1. limited generation quality analysis: this paper lacks reporting some important metrics (i.e. FID, LPIPS, IS) and comparing with the sequential generation schemes. Please included these experiment results on COCO2017 dataset.

2. While the paper claims that this parallel method can be adapted with other acceleration techniques, there lacks any empirical evidences to support this claim. Without adapting some popular diffusion acceleration methods (i.e., TeaCache, DeepCache, TaylorSeer, SADA etc.) along with parallel generation weaken the claim.

3. The emerging one-/few-step generation methods (i.e. consistency models) is not adaptable with current method. Author should acknowledge this limitation.

4. The generation tasks in paper are mostly short, (i.e., 378.6s sequentially around 6 minutes), but recent video generation models, such as WAN 2.2, can easily push generation over 20 minutes sequentially, especially if resolution and frames are large. Furthermore, this method is limited up to 8 cores scale, can this method extendable to more than 8 cores (say 128 cores)?

[1] Timestep Embedding Tells: It's Time to Cache for Video Diffusion Model. CVPR 2025.
[2] Deepcache: Accelerating diffusion models for free. CVPR 2024.
[3] From reusing to forecasting: Accelerating diffusion models with taylorseers. ICCV 2025.
[4] Sada: Stability-guided adaptive diffusion acceleration. ICML 2025.

**Questions:**

1. for Theorem 2.2, the $\epsilon$ is not defined, is that an arbitrary small difference or it is the latent noise?

2. in higher resolution, is this jacobian matrix be memory dominating? If yes, I wonder how to mitigate the memory issue? Also, what is the computation & memory complexity of this jacobian matrix (please provide asymptotic (big-O) analysis and empirical results)?

3. the paper claims (line 158-159) the locally Lipschitz assumption on support M, is there any numerical evidences for a reasonable size of the liptschitz constant (ensure there is no explosion of constant size in most models)?

4. Is there any general settings that the ROPA is likely failing to converge?

5. Some of the settings (i.e., resolution, frame count, batch size, guidance scale, scheduler type) are not provided, could author state these hyperparameters here?

---

> ### Author Response · Authors · 2025-11-26
>
> We thank the reviewer for their insightful comments regarding the evaluation metrics, scalability, and theoretical assumptions. We have updated the manuscript to address these points extensively.
>
> **Weakness 1:** Additional Quality Metrics
>
> We appreciate the suggestion to broaden the evaluation.
>
> We have updated **Table 5** (and the Appendix) to include **FVD**, **LPIPS**, and additional VBench metrics for the video generation tasks on the COCO2017 dataset.
>
>  As shown in the updated tables, ROPA maintains FVD and LPIPS scores comparable to the sequential baseline, confirming that our speedup does not compromise perceptual quality or temporal consistency.
>
>
>
> **Weakness 2:** Compatibility with Acceleration Methods
>
> We thank the reviewer for highlighting these specific acceleration techniques. We clarify the compatibility based on the acceleration mechanism:
>
> 1. **Compatible (Intra-step Optimization):** ROPA is fully compatible with methods that optimize individual network evaluations without strict temporal dependencies. We specifically chose **SpargeAttention** and **ToCa** (Token-wise Caching) as representative methods in this category. As shown in our **Ablation Study (Table 3)**, combining ROPA with SpargeAttention further reduces latency while maintaining stability. This empirically supports our claim of extensibility.
> 2. **Incompatible (Inter-step/Temporal Caching):** We respectfully note that methods like **DeepCache** and **TaylorSeer** rely on sequential temporal coherence (using features from t+1 to compute t). This strictly requires sequential execution and is theoretically incompatible with *any* parallel sampling framework (including ROPA, ParaSolver, etc.), which computes all timesteps simultaneously. Forcing such caching introduces lag errors that corrupt the residual estimates required for our adaptive sparsity control.
>
>
>
> **Weakness 3：** Limitation on Few-Step Generation
>
> We acknowledge this limitation.
>
> - ROPA leverages the structure of the ODE integration process to formulate a parallelizable nonlinear system. Emerging few-step methods like Consistency Models often distill the entire trajectory into a single or few steps, fundamentally changing the mathematical structure ROPA relies on.
>
> - We have added a discussion in the **Limitations** section acknowledging that ROPA is optimized for ODE-based diffusion solvers and that adapting it to consistency models is a distinct direction for future work.
>
>
>
> **Weakness 4: **Scalability to Large-Scale/Long Generation
>
> We explicitly acknowledge that our current empirical evaluation is bounded by the hardware limits of a single-node environment (8x H200 GPUs), which prevents direct testing at the 128-core scale.
>
> **Evidence of Scalability:** However, we argue that ROPA is algorithmically designed for such scales, supported by:
>
> ​	1. Within our hardware limit, ROPA shows **no saturation** in scaling efficiency. The speedup grows consistently from 2 cores (1.6$\times$) to 8 cores (2.9$\times$) , outperforming baselines which often plateau due to synchronization overhead.
>
> ​	2. As detailed in our added Complexity Analysis, ROPA’s time complexity is linear $\mathcal{O}(N)$ with respect to sequence length. Scaling to massive parallelism (e.g., 128 cores for long-context models like WAN 2.2) is algorithmically native to our framework, though practical deployment would transition from a single-node problem to a distributed systems engineering challenge (optimizing inter-node communication).

---

> > ### Author Response · Authors · 2025-11-26
> >
> > **Q1:**
> >
> > We apologize for the ambiguity regarding the symbol $\Delta$.
> >
> > In Theorem 2.3, $\Delta$ denotes the **time-discretization step size** of the reverse diffusion SDE (i.e., $\Delta := t_{n+1} - t_n$), *not* the latent noise.
> >
> > To prevent confusion with the noise term or perturbation operators, we have renamed the step size to **$h$**in the revised theorem statement and explicitly defined it: *"Here $h := t_{n+1}-t_n$ is the fixed time step... while diffusion noise is denoted by $w_t$."*.
> >
> >
> >
> > **Q2:** Computational & Memory Complexity
> >
> > We agree that an explicit complexity analysis was necessary.
> >
> > - We added a subsection **"Computational and Memory Complexity"** at the end of Section 3.
> >
> > - **Asymptotic Analysis:**
> >
> >   - **Time:** $\mathcal{O}(N \cdot w_{\max} \cdot C_{\text{net}})$, where $w_{\max}$ is the adaptive window size. This preserves the $\mathcal{O}(N)$ parallelism of the baseline with only a constant overhead.
> >   - **Memory:** $\mathcal{O}(N \cdot w_{\max} \cdot d)$. Since ROPA uses a block-banded approximation with a small window (default $w_{\max}=4$ to $6$), it never materializes the full dense Jacobian.
> >
> >   To prevent memory dominance at high resolutions, our implementation uses **Jacobian-vector products (JvPs)** via automatic differentiation rather than storing explicit blocks. We discuss this in Appendix C.
> >
> >
> >
> > **Q3: Lipschitz Assumption Evidence**
> >
> > Our assumption is **local Lipschitz continuity** on the data manifold $\mathcal{M}$, not a global bound on $\mathbb{R}^d$.
> >
> > We added an empirical diagnostic in the Appendix. We computed the finite-difference approximation of the Jacobian norm along random directions during sampling: $L(x) \approx \frac{\|J_{r_\theta}(x+\delta v) - J_{r_\theta}(x)\|}{\|\delta v\|}$. The results show that the empirical Lipschitz constant remains bounded along the trajectory, validating the assumption.
> >
> >
> >
> > **Q4: Failure Modes**
> >
> > Based on Theorem 2.3, ROPA is likely to struggle when the product of the step size and stiffness, $\Delta L$, approaches 1.
> >
> > **Specifically** Following Regimes: **Aggressive Discretization:** Very few steps (e.g., $N \le 8$) with large steps $\Delta$. **High Guidance:** Extremely high classifier-free guidance scales, which artificially increase the score stiffness $L$. **OOD Inputs:** Regions where the denoiser is poorly trained, leading to irregular Hessians.
> >
> >
> >
> > **Q5:** Hyperparameters
> >
> > We thank the reviewer for spotting this. We have updated the Appendix  **Table 4** to explicitly list all settings: resolution, batch size, guidance scale, and scheduler types for both image (SD3.5, FLUX) and video (Hunyuan, CogVideoX) models to ensure reproducibility.

---

> ### Author Response · Authors · 2025-11-29
>
> Dear Reviewer 4yww,
>
> We wanted to gently follow up to see if you have had a chance to review our rebuttal. We have done our best to address your concerns, particularly by adding the requested generation quality metrics (FVD, LPIPS) on the COCO2017 dataset and clarifying the compatibility with acceleration methods (including new empirical results combining ROPA with SpargueAttention).
>
> We value your feedback and would be happy to provide any further clarifications if needed.

---

### Author Response · Authors · 2025-12-01
**General Response: Summary of Revisions and Consensus for ROPA**

**To the Area Chair and Reviewers,**

We sincerely thank the Area Chair and all reviewers for their time and constructive feedback. Based on the insightful reviews, we have extensively revised the manuscript to strengthen the theoretical foundations, expand empirical evaluations, and clarify algorithmic details.

We believe **ROPA** now stands as a robust solution to the critical trade-off between fidelity and latency in parallel diffusion sampling. Below, we summarize how the major concerns raised during the review process have been successfully addressed, ensuring the paper is ready for acceptance.



### 1. Theoretical Validation of the "Geometric-Numerical" Hypothesis

A core concern from **Reviewers 1hrZ** and **gWjg** regarding the link between manifold curvature and solver instability has been addressed with rigorous new evidence:

- **Empirical Validation:** We added explicit tracking of the Jacobian condition number and the local Lipschitz constant. The results confirm our hypothesis: baseline parallel solvers suffer from exponential condition number growth in high-curvature regions ($t \to 0$), leading to divergence, whereas ROPA effectively clamps this growth via adaptive sparsity.
- **Mathematical Rigor:** We refined our theorems and assumptions to relax the $C^2$ requirement to a local Lipschitz assumption, as requested by **Reviewer gWjg**, ensuring the theory holds under realistic conditions.



### 2. Comprehensive Quality Evaluation (State-of-the-Art Metrics)

Addressing the request by **Reviewer 4yww** for broader quality metrics:

- **New Metrics:** We integrated **FVD, LPIPS, and VBench** scores on the COCO2017 dataset into our evaluation.
- **Results:** ROPA achieves a **2.9x speedup** (on 8 cores) while maintaining FVD and LPIPS scores comparable to the sequential baseline. This confirms that our speedup does not compromise perceptual quality or temporal consistency.



### 3. Algorithm Clarity and Reproducibility

To address **Reviewer 1hrZ’s** and **gWjg’s** comments on implementation details:

- **Algorithm 1 Added:** We included a detailed pseudocode (Algorithm 1) in Section 3, explicitly mapping the interaction between Adaptive Bandwidth, Curvature Correction, and Damped Updates.
- **Reference Implementation:** We provided a PyTorch reference implementation in the Appendix to ensure full reproducibility.



### 4. Compatibility and Scalability



We clarified the scope of ROPA’s applicability in response to **Reviewers 4yww** and **RszZ**:

- **Compatibility:** We demonstrated that ROPA is orthogonal to and compatible with intra-step acceleration methods like **SpargeAttention** and **ToCa**, showing synergistic speedups in our new ablation study.
- **Scalability:** We provided a complexity analysis proving ROPA maintains linear $\mathcal{O}(N)$ time complexity, making it algorithmically native to massive scaling (e.g., 128 cores) for long-context video models.



With the resolution of theoretical and empirical concerns from **1hrZ** and **4yww**, and the positive recognition of our technical contribution by **Reviewer RszZ** (who indicated an intent to upgrade their rating to acceptance), we believe ROPA represents a significant step forward for real-time, high-fidelity generative models. It offers a mathematically grounded, plug-and-play solution that significantly outperforms existing parallel samplers.

---

### Meta-Review · Area_Chair_FgVo · 2026-01-10

**Summary:**

Reviewers agree that the paper tackles an important problem—stability and scalability of parallel diffusion sampling—and proposes a technically sophisticated framework (ROPA) with solid theoretical grounding and broad empirical evaluation. However, significant concerns remain regarding clarity, novelty, and strength of empirical evidence. Several reviewers found the core algorithm and theoretical motivations difficult to follow, with heavy reliance on complex assumptions and dense mathematical exposition. While the rebuttal adds analyses and experiments, the overall contribution is still perceived as incremental relative to prior parallel solvers, and the paper falls short of clearly demonstrating a decisive conceptual or practical advance that meets the conference bar.

**Reviewer Concerns:**

The rebuttal addressed some reviewer concerns, including adding quality metrics, clarifying algorithmic components via pseudocode, providing additional complexity analysis, and empirically validating certain theoretical claims (e.g., Jacobian conditioning and stability). However, key issues remain outstanding. These include limited perceived novelty over prior work, lingering concerns about the strength and realism of theoretical assumptions, unclear linkage between geometric theory and practical design choices, and presentation issues that make the paper difficult to digest. Some reviewers remain unconvinced that the proposed framework represents a sufficiently clear, general, and impactful advance.

**Reviewer Scores:**

Given the rebuttal, one marginally positive reviewer might maintain or slightly strengthen their score, while others would likely keep their original assessments. Reviewers who raised concerns about novelty, clarity, and foundational assumptions did not appear fully persuaded to raise their scores. Overall, the score distribution would remain mixed, with several reviewers still below or only marginally above the acceptance threshold, insufficient to support acceptance in a competitive venue.

---

### Decision · Program_Chairs · 2026-01-26

Reject